# Frequency Estimation from Compressed Measurements of a Sinusoid in Moving-Average Colored Noise

Nuha A. S. Alwan [1] and Zahir M. Hussain [2,*]

1   College of Engineering, University of Baghdad, Baghdad 10011, Iraq; n.alwan@ieee.org
2   School of Engineering, Edith Cowan University, Joondalup 6027, Australia
*   Correspondence: z.hussain@ecu.edu.au

**Abstract:** Frequency estimation of a single sinusoid in colored noise has received a considerable amount of attention in the research community. Taking into account the recent emergence and advances in compressive covariance sensing (CCS), the aim of this work is to combine the two disciplines by studying the effects of compressed measurements of a single sinusoid in moving-average colored noise on its frequency estimation accuracy. CCS techniques can recover the second-order statistics of the original uncompressed signal from the compressed measurements, thereby enabling correlation-based frequency estimation of single tones in colored noise using higher order lags. Acceptable accuracy is achieved for moderate compression ratios and for a sufficiently large number of available compressed signal samples. It is expected that the proposed method would be advantageous in applications involving resource-limited systems such as wireless sensor networks.

**Keywords:** frequency estimation; compressive covariance sensing; linear sparse ruler; least squares; colored noise





## 1. Introduction

Frequency estimation of a sinusoid in noise is one of the major problems in signal processing and communications with various applications in radar, sonar, carrier synchronization and signal detection, among many others. Many frequency estimation methods have been devised for the white noise case such as the maximum likelihood (ML) estimation method, which involves locating the peak of the periodogram [1]. This efficient method attains the Cramer–Rao lower bound (CRLB) asymptotically but is computationally complex even when the fast Fourier transform (FFT) is used. In addition, a number of fast and accurate frequency estimators of a sinusoid in white noise have been developed as linear prediction estimators, also achieving the CRLB on variance asymptotically [2]. A more computationally efficient method than the ML method is the correlation method [3,4] in which an estimate of the frequency is obtained from knowledge of one or more values of the sinusoid-plus-noise autocorrelation sequence. Improved-performance frequency estimators have been obtained using multiple autocorrelation lags, such as the P-estimator [5] and the modified Pisarenko harmonic decomposer (MPHD) [6]. However, they involve phase unwrapping to overcome the resulting frequency ambiguities, thereby increasing complexity. Digital phase locked loops (DPLL) offer low-complexity, sample-by-sample frequency estimation for real-time applications in contrast to the above-mentioned batch estimators [7–9]. Disadvantages of DPLL estimators are overshoot and long settling times [10]. In [11], unbiased estimation of a sinusoid in colored noise is achieved via adapted notch filters. Colored noise is practically more relevant and appears in many applications [12]. A more realistic description of noise in electronic devices and systems for instance is given by correlated or colored noise. White noise is rarely an appropriate model to represent noise sources in such systems as electronic sinusoidal oscillators [13].

The purpose of the present work is to perform single-tone frequency estimation in colored noise using correlation-based methods, as they allow frequency estimation from

compressed measurements using compressive covariance sensing (CCS). CCS is akin to compressive sensing (CS) [14]. As asserted in [15], CS has great potential for energy-efficient data processing and communication in the wireless sensor networks (WSN) that have become commonplace due to their major role is various applications, such as industrial monitoring, healthcare monitoring environmental sensing etc. CS compresses data by sampling randomly at a sub-Nyquist rate to gain numerous compression advantages, such as limitation of sensing, storage and communication costs, thereby making CS ideal for resource-limited systems such as WSNs. The original uncompressed signal can be recovered from its compressed version on the condition that the signal is sparse in a certain transform domain. In contrast, CCS techniques [16] can recover the second-order statistics of an original signal that has been compressed rather than the signal itself without the exertion of the sparsity condition and, at the same time, retaining all of the above-mentioned advantages of compression. CCS relies on certain structural forms of the covariance matrix to be captured during compression, enabling the recovery of covariance (or correlation) information of wide sense stationary (WSS) signals and, even more recently, nonstationary signals via online CCS [17]. One of the major applications of CCS is correlation-based frequency estimation from compressed measurements. Other applications of CCS include power spectrum estimation [18,19] and direction-of-arrival estimation in array signal processing [20].

Frequency estimation of a sinusoid from compressed measurements given the additive white Gaussian noise (AWGN) case has been addressed in the literature [21–24], and frequency estimation of more than one harmonic in a frequency-sparse signal is dealt with in [25]. In [21], ML estimation by grid search optimization is proposed such that the frequency estimate is that which maximizes a cost function. In [22], a Newton-like algorithm is used to further refine the estimate after grid search, whereas the ML estimator in [23] requires a course-fine grid search. In [24,26], frequency estimation from compressed measurements is cast as a linear least squares (LS) problem. The work in [24] is concerned with frequency estimation of a sinusoid in white noise undergoing compression. The compression or sampling matrix is taken as a random matrix rather than a sparse matrix of zeros and ones. Therefore, this matrix colors the white noise during compression. As will be explained in the forthcoming sections, our approach is different in that the contaminating noise is itself assumed to be colored, while the sampling matrix is sparse and does not further color the noise samples upon compression. Another difference is the use of CCS in the present work due to adopting the correlation-based frequency estimation approach.

Correlation-based frequency estimation of a real-valued single tone in colored noise has been dealt with in [27] using higher-order lags, thereby restricting the treatment to moving-average (MA) noise and achieving low complexity. The MA filter is the most commonly used filter in digital signal processing due to its simplicity and understandability [28]. Therefore, there are undoubtedly many instances in which the frequency of a noisy sinusoid must be estimated at the outputs of such filters that cause coloring of the contaminating white noise. We focus on extending the work of [27] to achieve frequency estimation from compressed sinusoids in MA colored noise using CCS. This approach in the compressed measurements context has the advantage of reduced computational complexity in comparison with the ML approaches of [21–24] and even with the correlation-based approaches of [5,6] since no phase unwrapping is involved, as well as the applicability to the colored noise case. In brief, the contribution of the present work lies in applying the existing novel technology of CCS to estimate the frequency of a sinusoid in MA colored noise from compressed measurements by correlation-based methods, thereby gaining a computational efficiency advantage over relevant ML approaches [21–24] in the compressed-measurement context that rely on computationally complex grid search optimization. Moreover, and to the best of the authors' knowledge, there are no studies in this direction in the literature that deal with the MA colored noise case except the work in [27] that does not consider compressed measurements.

Recently, the emerging artificial intelligence technique of deep learning (DL) found application in frequency estimation of noisy sinusoidal signals [29,30] with promising performance. However, no results were reported for frequency estimation of compressed sinusoids using DL; this remains a topic for future research.

The remainder of the paper is organized as follows. Section 2 summarizes the frequency estimation method of [27]. CCS is explained in Section 3 relating to the present frequency estimation problem. Simulation results are given in Section 4, and, finally, Section 5 concludes the paper.

## 2. Frequency Estimator of Real-Valued Single Tone in Colored Noise Using Multiple Autocorrelation Lags

The estimator proposed in [27] can be summarized as follows. A real-valued sine wave $s(n)$ of amplitude $\alpha$, frequency $\omega_o$ and phase $\varphi$ is immersed in colored noise $u(n)$ resulting in the length-$L$ observed signal samples $v(n)$ given by:

$$v(n) = s(n) + u(n) = \alpha \sin(\omega_o\, n + \varphi) + u(n), \qquad 1 \leq n \leq L \qquad (1)$$

The digital frequency $\omega_o$ measured in radians is equal to $2\pi\, f_o/f_s$, where $f_o$ is the sinusoid analog frequency in Hz and $f_s$ is the sampling frequency in Hz. The colored noise $u(n)$ is zero-mean, WSS, Gaussian and independent of $s(n)$. It is assumed to be generated by passing zero-mean AWGN of variance $\sigma^2$ through an order-$Q$ finite impulse response (FIR) filter. The normalized noise autocorrelation

$$\rho(k) = \{E[u(n)\, u(n-k)]\}/\sigma^2$$

is equal to zero for $k > Q$. The signal to noise ratio (SNR) is taken as $\alpha^2/2\sigma^2$.

The autocorrelation function of $v(n)$ is denoted by $r(k)$ and is given by:

$$r(k) = [v(n)v(n-k)] = \frac{\alpha^2}{2} \cos(k\, \omega_o) + \sigma^2\, \rho(k) \qquad (2)$$

where  is the expectation operator.

Using trigonometric relations, it is straightforward to show that an estimate of the frequency $\omega_o$, denoted by $\hat{\omega}_o$, can be expressed as

$$\hat{\omega}_o = \cos^{-1}\left[\frac{\sum_{k=p}^{q} \hat{r}(k)\, [\hat{r}(k-1) + \hat{r}(k+1)]}{2\, \sum_{k=p}^{q} \hat{r}(k)^2}\right] \qquad (3)$$

for any integer $q > p > Q + 1$, and $\hat{r}(k)$ is the unbiased estimation of autocorrelation given by

$$\hat{r}(k) = \frac{1}{L-k} \sum_{n=1}^{L-k} v(n)\, v(n-k) \qquad (4)$$

This method of single-tone frequency estimation has been shown in [27] to outperform the P-estimator and the MPHD methods in terms of computational load and accuracy and for the colored noise case. We next discuss the application of the CCS method to the frequency estimator discussed above while considering compressed measurements of the noisy sinusoid.

## 3. The Compressive Covariance Sensing Method and Its Application to Frequency Estimation

Autocovariance of a signal is equivalent to its autocorrelation when the signal mean is zero [31], and it will be referred to throughout the remainder of this paper as covariance for simplicity. CCS recovers the covariance of a signal from a compressed version of it when the compression is carried out by further sampling the signal using a linear sparse ruler (LSR) [16]. An LSR with a length of $N-1$, $N = 11$, is shown in Figure 1. The $M = 6$ marks on

the ruler defined by the set {0,1,3,7,8,10} allow all integer distances between zero and 10 to be measured, which can be easily verified.

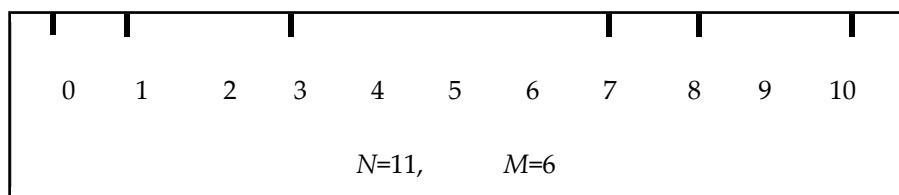

**Figure 1.** A linear sparse ruler of length 10. The set of marks is {0,1,3,7,8,10}.

The compression ratio is $M/N$. A minimal LSR has the least number of marks $M$ and hence the best compression ratio. Since all possible lags can be defined, the covariance sequence of the original signal can be recovered from the compressed signal.

Covariance matrices are symmetrical and Toeplitz (ST) for real signals [16,31,32]. The covariance sequence constitutes the first column of the recovered covariance matrix.

The relation between a signal vector $\mathbf{x} \in \mathbf{R}^N$ and its compressed version $\mathbf{y} \in \mathbf{R}^M$ is

$$\mathbf{y} = \Phi\,\mathbf{x} \tag{5}$$

where $\Phi = \mathbf{R}^{M \times N}$ is a sparse sampling matrix that performs the function of the LSR. Each of the $M$ rows of $\Phi$ has at most one non-zero (unity) value.

The theoretical covariance matrix of vector $\mathbf{x}$ will be given the symbol $\Sigma$ for convenience and is described by

$$\Sigma = \left[\mathbf{x}\,\mathbf{x}^T\right] \tag{6}$$

where the superscript $T$ denotes transpose. The above matrix can be thought of as a linear combination of ST matrices that are elements of the subspace

$$\Im = \{\Sigma_o,\,\Sigma_1,\,\ldots,\,\Sigma_{S-1}\} \subset \mathbf{R}^{N \times N}.$$

There exist real scalars $\alpha_s$ such that

$$\Sigma = \sum_{s=0}^{S-1} \alpha_s\,\Sigma_s \tag{7}$$

When the subspace $\Im$ and therefore the matrices $\Sigma_s$ are chosen, knowledge of the $\alpha_s$ leads to knowledge of $\Sigma$. The number $S$ of the matrices $\Sigma_s$ is restricted to $S < 2N - 1$ [32].

The covariance matrix of the compressed vector is

$$\overline{\Sigma} = \left[\mathbf{y}\,\mathbf{y}^T\right] \tag{8}$$

Substituting Equation (5) and Equation (6) into Equation (8) yields:

$$\overline{\Sigma} = \Phi\,\Sigma\,\Phi^T = \sum_{s=0}^{S-1} \alpha_s\,\overline{\Sigma}_s \tag{9}$$

where

$$\overline{\Sigma}_s = \Phi\,\Sigma_s\,\Phi^T \tag{10}$$

The matrix $\overline{\Sigma}$ is a linear combination of the symmetric matrices that are elements of the subspace $\overline{\Im} = \{\overline{\Sigma}_o,\,\overline{\Sigma}_1,\,\ldots,\,\overline{\Sigma}_{S-1}\} \subset \mathbf{R}^{M \times M}$. They are not necessarily Toeplitz. If compression is achieved using an LSR and thereby preserving the second-order statistics, then the subspace $\overline{\Im}$ is linearly independent, and knowing $\overline{\Sigma}$ from Equation (8) and $\overline{\Sigma}_s$ from Equation (10) leads to knowing the $\alpha_s$ from Equation (9); finally, $\Sigma$ is found from Equation (7).

The computation of the $\alpha_s$ from Equation (9) leads to an overdetermined system of equations that can be solved by LS [16,32]. We also note that the covariance of the compressed vector given in Equation (8) must be estimated to solve the CCS problem. In [33], it was proved that using the unbiased covariance estimation method of Equation (4) for least squares CCS (LS-CCS) gives the best results.

To apply CCS to our frequency estimation problem, the length-*L* sequence of noisy sinusoidal samples of Equation (1) is divided into *B* blocks of length *N*. Each block is compressed by the LSR of compression ratio *M/N*, and CCS is carried out. The number *S* of linearly independent matrices $\Sigma_s$ in the subspace $\Im$ is taken as equal to *N*, and the subspace is chosen as

$$\Im = \{I_N\} \cup \{T_1, T_2, \ldots\ldots, T_{N-1}\} \tag{11}$$

where $I_N$ is the identity matrix and $T_k$ is the Toeplitz matrix with ones on the diagonals *k* and $-k$ and zeros elsewhere. The final recovered covariance values are the result of averaging over the *B* blocks [16,34]. These values are used in Equation (3) to estimate the frequency. The algorithmic steps of the whole procedure are outlined below.

---

**Algorithm 1: CCS-based frequency estimation of a compressed noisy sinusoid.**

---

**Input**: A length-*L* noisy sinusoid divided into *B* blocks of length *N*. Each of the *B* blocks undergoes compression by an LSR with compression ratio *M/N*. Denoting the corresponding sampling matrix performing the function of the LSR by $\Phi$, we have y $= \Phi$ x, where y is the length-*M* compression of the length-*N* block represented by x.

a.    For each compressed block, perform the following:

      1.    Choose *S* linearly independent ST $N \times N$ matrices $\Sigma_s$ using the subspace of Equation (11).

      2.    For each $\Sigma_s$, find an $M \times M$ matrix $\overline{\Sigma}_s$ from Equation (10): $\overline{\Sigma}_s = \Phi \, \Sigma_s \, \Phi^T$.

      3.    Compute an unbiased estimate of the ST $M \times M$ compressed signal covariance matrix $\overline{\Sigma}$ using the following equation that yields the first row of $\overline{\Sigma}$ whose elements are denoted by $\hat{r}_y(k)$:

$$\hat{r}_y(k) = \frac{1}{M-k} \sum_{n=1}^{M-k} y(n)\, y(n-k), \qquad k = 0, 1, \ldots, M-1$$

       .

      4.    Find the $\alpha_s$ by LS from Equation (9): $\overline{\Sigma} = \sum_{s=0}^{S-1} \alpha_s \, \overline{\Sigma}_s$.

      5.    Find $\hat{\Sigma}_{LS}$, which is the LS estimate of the ST covariance matrix $\Sigma$ of the original length-*N* block from Equation (7): $\hat{\Sigma}_{LS} = \sum_{s=0}^{S-1} \alpha_s \, \Sigma_s$.

b.    Repeat for all *B* blocks and average the estimated covariance over all blocks. Denote the first row of the estimated covariance matrix by $\hat{r}(k)$, $k = 0, 1, \ldots, N-1$.

c.    Find the estimated frequency by applying Equation (3):

$$\hat{\omega}_o = \cos^{-1}\left[ \frac{\sum_{k=p}^{q} \hat{r}(k)\, [\hat{r}(k-1) + \hat{r}(k+1)]}{2 \sum_{k=p}^{q} \hat{r}(k)^2} \right],$$

    where $q > p > Q + 1$ and *Q* is the order of the FIR filter generating the colored noise.

**Output**: Frequency estimate $\hat{\omega}_o$ of the length-*L* noisy sinusoid.

---

## 4. Simulation Results

Simulations are carried out in MATLAB. The frequency estimator of [27] is first simulated. The length *L* of the real-valued noisy sinusoid is first chosen to be $L = 110$ samples. In Equation (1), we set $\alpha = 5$, $\omega_o = \pi/8$, and $\varphi = 0$ without loss of generality. The noise variance $\sigma^2$ is found from the SNR equation in dBs:

$$\text{SNR(dB)} = 10 \, \log_{10}\left( \frac{\alpha^2}{2\sigma^2} \right) \tag{12}$$

Zero-mean AWGN of variance $\sigma^2$ is passed through a FIR filter of order $Q = 2$, having a system transfer function $H(z) = (1/3)(1 + z^{-1} + z^{-2})$ to form the MA noise component $u(n)$ in Equation (1). Equations (3) and (4) are applied to obtain the frequency estimate denoted by $\hat{\omega}_o$, taking $p = 5$ and $q = 10$. The estimation means square error (MSE) is given in dBs by

$$\text{MSE} = 10\log_{10}\left[\left\{(\omega_o - \hat{\omega}_o)^2\right\}\right]. \tag{13}$$

To make a meaningful plot of estimation of MSE in dB vs. SNR (dB), we need to plot the CRLB as well. The CRLB provides the minimum variance of the unbiased estimator as SNR increases. For the problem of frequency estimation of a single sine in white noise, the CRLB was found in [35] to be:

$$\text{CRLB} = \frac{12\,\sigma^2}{\alpha^2\,L\,(L^2 - 1)} \tag{14}$$

where $L$ is the number of signal samples. To take into account MA colored noise [36], we take into account the noise spectral density at the sinusoidal position $\omega_o$. The expression becomes

$$\text{CRLB} = \frac{12\,\sigma^2\left|H\left(e^{j\,\omega_o}\right)\right|^2}{\alpha^2\,L\,(L^2 - 1)} \tag{15}$$

where $H\left(e^{j\omega_o}\right) = H(z)\big|_{z = e^{j\omega_o}}$.

The above holds provided that the noise spectrum is relatively smooth over frequency intervals corresponding to $2\pi$ times the reciprocal of the number of signal samples [36], which is the case in the present settings. The approximation improves as the number of samples increases.

Next, a compressed version of the length—110 noisy sinusoid is considered by first dividing the signal vector into 10 blocks ($B = 10$), each with a length of $N = 11$. Each block is compressed using an LSR with a compression ratio $M/N = 6/11$ as in Figure 1. CCS is then applied as explained in Algorithm 1 of Section 3 to the compressed vector to recover the correlation values to be substituted in Equation (3). The process is repeated using $M/N = 9/11$ for which the LSR is {0,1,3,4,6,7,8,9,10}.

All simulations are averaged over 500 independent runs. The results of the above procedure are demonstrated in Figure 2. It is a plot of the MSE versus SNR for the estimator with no compression corresponding to [27], the estimator of the compressed noisy sinusoid using CCS with compression ratio 9/11 and finally its 6/11 counterpart. The CRLB is also shown. Clearly, as compression increases, performance is degraded. Figure 3 demonstrates the MSE versus SNR for only one compression ratio (9/11) but for different values of the number of blocks $B$. The improvement is clear as $B$ increases. Increasing $B$ may appear to be contradictory to the concept of compression, but in fact, it is not since the average sampling rate is reduced by compression regardless of the length $L$ ($=B \cdot N$) of the signal, and it is the sampling rate that determines hardware complexity and consequently cost-effectiveness [16]. For the case of compression ratios equal to 8/11 and 9/11, the estimation is acceptable as can be discerned from Figure 4, which shows the estimated frequency $\hat{\omega}_o$ versus SNR, although these LSRs are not minimal. The LSR with a compression ratio of 8/11 was implemented with the set {0, 1, 3, 5, 6, 7, 8, 10}. The compression ratio 8/11 also results in an estimation bias of 0.038 rad (or 0.006 times the sampling frequency). This is clear from Figure 4. The advantages of compression may outweigh the disadvantage of reduced accuracy depending on the application.

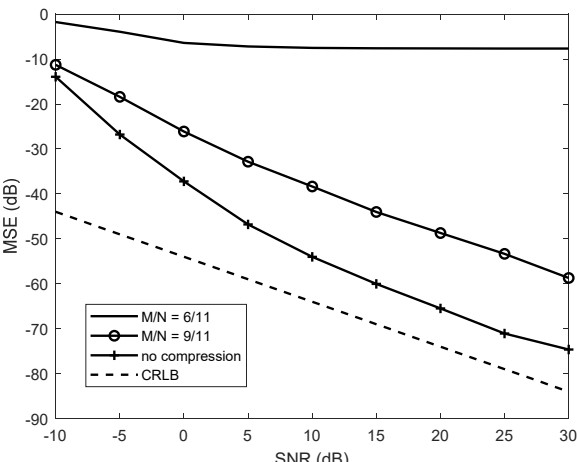

**Figure 2.** Frequency estimation mean square error versus SNR, the colored noise case, using CCS with different compression ratios, as well as the estimator without compression [27]. $\omega_o = \pi/8$ rad, $\alpha = 5$, $B = 10$, $p = 5$, $q = 10$.

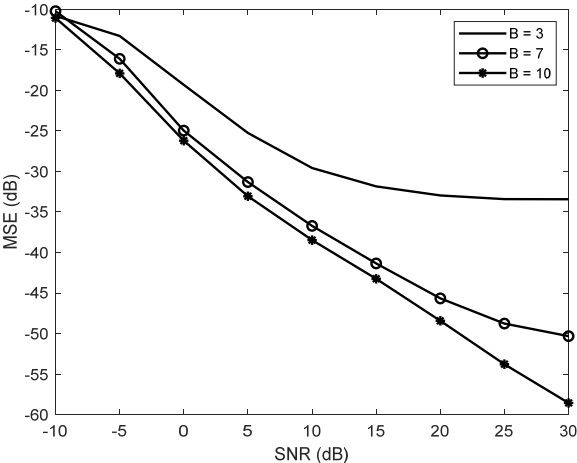

**Figure 3.** Frequency estimation mean square error versus SNR, the colored noise case, using CCS with compression ratio $M/N{=}9/11$ and for different values of $B$ (number of blocks). $\omega_o = \pi/8$ rad, $\alpha = 5$, $B = 10$, $p = 5$, $q = 10$.

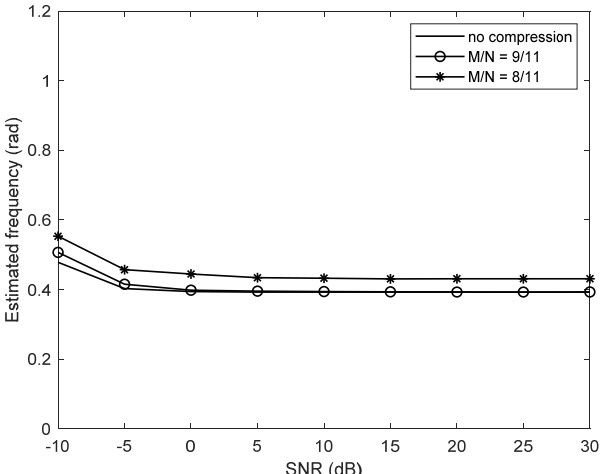

**Figure 4.** Frequency estimate ($\hat{\omega}_o$) versus SNR, the colored noise case, using CCS with different compression ratios as well as the estimator without compression [27]. $\omega_o = \frac{\pi}{8} = 0.393$ rad, $\alpha = 5$, $B = 10$, $p = 5$, $q = 10$.

The frequency estimation performance increases with higher compression ratios (less compression). Figure 5 is a plot of the frequency estimation mean square error versus the compression ratio $M/N$ for two colored noise scenarios: SNR = 5 dB and SNR = 30 dB. The compression ratios involved are 6/11, 7/11, 8/11, 9/11 and unity (no compression). The 6/11 minimal LSR is used as in Figure 1, and then the higher compression ratios are implemented by adding marks to the ruler at random. This method is used in [18] to achieve higher compression ratios. Figure 5 clearly shows the increase in mean square error with stronger compression. It can be seen that for low compression ratios, the effect of SNR is negligible since the estimation is biased as previously demonstrated in Figure 4, even for high SNR.

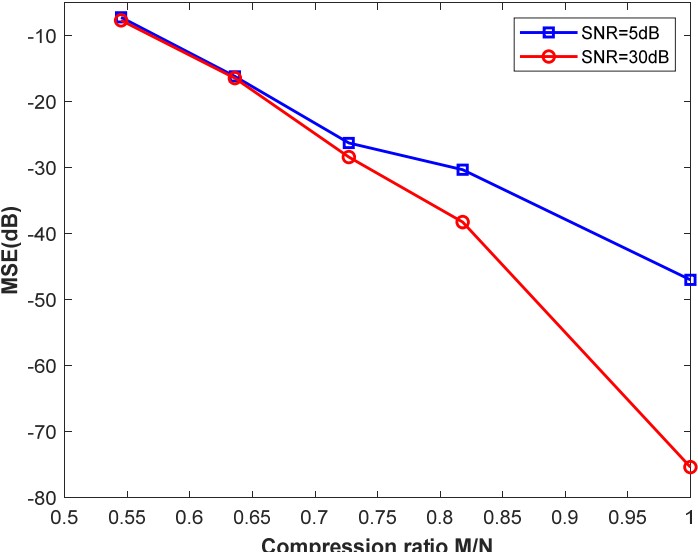

**Figure 5.** Frequency estimation mean square error versus compression ratio, the colored noise case, for SNR = 5 dB and SNR = 30 dB. $\omega_o = \pi/8 = 0.393$ rad, $\alpha = 5$, $B = 10$, $p = 5$, $q = 10$.

Finally, the frequency estimation MSE as a function of $\omega_o$ for the white noise case and the colored noise case is shown in Figures 6 and 7, respectively, and for SNR = 30 dB. In Figure 6, the values of $p$ and $q$ are taken as 2 and 10, respectively. Increasing the number of the CCS-recovered autocorrelation lags results in better performance. The no-compression case clearly approaches the CRLB. Performance is compromised as expected when the required compression operation is performed. Figure 7 is a repetition of Figure 6 for the colored noise case. However, here, the value of $p$ is taken as 5 and $q$ as 10, since $p$ must be greater than $Q+1$, where $Q$ is the order of the coloring filter previously described. In this case, the CRLB is a function of $\omega_o$ as can be verified by Equation (15). This is in contrast to the constant CRLB in the white noise case. The performance is comparable to that of Figure 6, but compression slightly degrades the MSE in Figure 7 regarding the colored noise case. It can be observed that the 9/11 compression results are better (lower MSE) for the white noise case due to using more CCS-recovered autocorrelation lags in the estimation process. In both figures, the compression results are perceptibly better in the range of $\omega_o$ less than $(\pi/2)$ rad.

The CCS algorithm itself is not directly related to colored noise, but the work in [27] on which the present work is based is MA colored noise resilient. The CCS method has been incorporated in this work to compute the correlation values for the case of compressed measurements. The work in [27] and, consequently, the present work are resilient to MA noise because, as stated in Section 2, the autocorrelation of the colored noise is zero for shifts greater than the order of the MA stochastic process of the colored noise. Figures 6 and 7 support this argument since the performances in white and colored noise are comparable, although the white noise case shows a slightly better performance due

to using more correlation lags than in the colored noise case. The only restriction in the colored noise case is that the lags must be greater than the MA order; therefore, there are fewer available lags to take part in the estimation process.

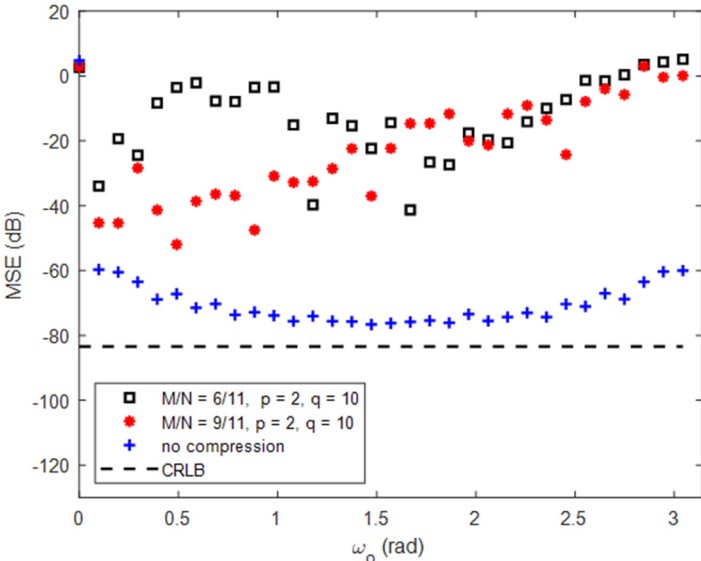

**Figure 6.** Frequency estimation mean square error versus $\omega_o$ for the white noise case. SNR = 30 dB, $\alpha = 5$ and $B = 10$, using CCS with different compression ratios as well as the estimator without compression [27], $p = 2$, $q = 10$.

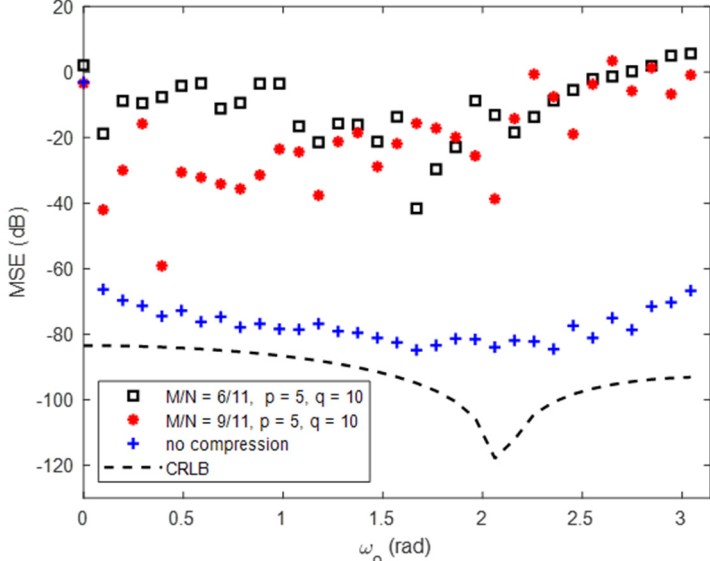

**Figure 7.** Frequency estimation mean square error versus $\omega_o$ for the colored noise case. SNR = 30 dB, $\alpha = 5$ and $B = 10$, using CCS with different compression ratios as well as the estimator without compression [27], $p = 5$, $q = 10$.

## 5. Conclusions

Compressed measurements of a single sinusoid in MA colored noise are used to reconstruct the second-order statistics for the purpose of frequency estimation of the original uncompressed noisy sinusoid using compressive covariance sensing. Estimation accuracy is acceptable for moderate compression ratios and was found to degrade for lower (better) compression ratios generally resulting in biased estimates. For a fixed compression ratio, the mean square estimation error was found to improve (decrease) as the number of available compressed signal samples increased. The best performance

results with compression were observed for sinusoid frequencies less than ($\pi/2$) radians. The advantages gained by compression may outweigh the disadvantage of decreased estimation accuracy such that the performance acceptability is based on a complexity-accuracy tradeoff that is application-dependent.

**Author Contributions:** Conceptualization, N.A.S.A. and Z.M.H.; methodology, N.A.S.A.; software, N.A.S.A.; formal analysis, N.A.S.A. and Z.M.H.; investigation, N.A.S.A. and Z.M.H.; resources, N.A.S.A. and Z.M.H.; writing—original draft preparation, N.A.S.A.; writing—review and editing, Z.M.H.; supervision, Z.M.H. All authors have read and agreed to the published version of the manuscript.

**Funding:** This research is partially funded by Edith Cowan University via the ASPIRE Program.

**Data Availability Statement:** The computer simulation MATLAB code is available from the authors on request.

**Acknowledgments:** The authors would like to thank Edith Cowan University for supporting this project via the ASPIRE Program. Thanks to the reviewers for their insightful comments that helped improve this paper. Thanks are extended to MDPI Office for their prompt attention and useful guidance throughout the review process.

**Conflicts of Interest:** The authors state that there is no conflict of interest in the publication of this work.

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
