# Peer review of "Frequency Estimation from Compressed Measurements of a Sinusoid in Moving-Average Colored Noise"

_electronics, doi:10.3390/electronics10151852_

Round 1

Reviewer 1 Report

The manuscript “Frequency estimation from compressed measurements of a sinusoid in MA colored noise” focuses on applying a compressive covariance sensing (CCS) technique in frequency estimation of harmonic signal occurred in presence of moving averaging (MA) colored noise. In my opinion, this topic is current and crucial, and the manuscript is interesting and prepared at a good technical level. However, the author should improve it before its publishing in the Electronics journal. Below, I showed some suggestions and comments, which could help to correct it.

Main disadvantages of reviewed manuscript:

  1. Manuscript title

In general, most journals do not recommend abbreviations in their titles. I suggest you do not use the MA abbreviation in the title. This abbreviation is not as common as a significant number of readers will not understand this abbreviation.

  1. Introduction & survey of literature

Generally, Sections 1 and 2 are written interesting. The authors presented a good literature review in the field of frequency estimation methods in sinusoidal signals. However, the introduction should be supplemented with a clear definition of the authors' contribution to the analyzed topic. The authors should also indicate the novelty and originality of the proposed approach.

  1. Abbreviations

Generally, the Author uses abbreviations correctly, i.e., introduces them (i.e., gives full name and abbreviation in brackets) separately for the abstract and the rest of the article. However, abbreviations should be used mainly when they are repeated often. Hence, the MA abbreviation does not have to be introduced in the abstract because it is not used in the rest of it. On the other hand, the abbreviations should only be introduced once the first time the full name appears, and then only the abbreviation should be used until the end of the article. Therefore, the full name or the full name with the abbreviation together should not be reintroduced in the remainder of the manuscript. This is an acceptable exception. However, the full name should not be used together with the abbreviation! For example, the FIR abbreviation (page 3, line 99) is not explained. The least-squares full name is occurred in lines 77 (page 2) and 168 (page 5), while the LS abbreviation is introduced in line 171 (page 5). The WGN abbreviation is introduced in line 217 (page 7), while ‘white Gaussian noise’ full name is already occurred in line 98 (page 3). I propose to replace them with the AWGN abbreviation introduced in line 66 (page 2). The MSE abbreviation is introduced twice in lines 221 and 223 (page 7). Please check out the full article and remove all redundancy.

  1. Proposed solution. Sections 2–4

The use of the frequency concept in relation to the phase angle, ωo, is not understandable to me. The frequency concept is inextricably linked with the frequency unit, i.e., herz. In Equation (1), there should be:

v(n) = s(n) + u(n) = α sin(2πfon/fs + φ) + u(n) = α sin(ωon + φ) + u(n)   (1)

where fo and fs  are the signal carrier frequency and sample rate, respectively, and ωo = 2πfo/fs is the phase angle that represents the current signal frequency.

On the other hand, the authors should justify the application of the proposed solution. Analyzing the results, I conclude that it is better to use 'no compression' approach because MSEs are smaller. So what is the purpose of using the approach with compression? This may be unclear to potential readers.

Figures 2-4 show average MSE values obtained based on 500 runs. It is worth adding the standard deviations of MSE to the presented results in the charts.

In simulation studies, please consider whether adding the MSE versus SNR analysis for the fixed parameters when changing the ωo value would be of value. If the obtained results would not depend on ωo, this idea can be ignored.

The simulation results should be presented in Section 4 instead of Section 5. Figures should be introduced right after the end of the paragraph, where they are referenced in the text.

  1. Text formatting and journal template

Please see the instruction for authors and template for the Electronics journal to improve the formatting of the manuscript – see links below or other papers published in the Electronics.

https://www.mdpi.com/journal/electronics/instructions

https://www.mdpi.com/files/word-templates/electronics-template.dot

Please correct the text formatting according to the template. Note in the template that all lines of text, including captions of figures and tables, should be numbered. Only equations, figures, and tables are unnumbered!

The authors should pay also attention to, e.g.,

  1. The article title should be written using capital letters, except for adverbs and articles, etc. Do not use hyphenation in the title.
  2. All equations and symbols in the text should be in the same font as the text, i.e., Palatino Linotype, size 10.
  3. Apply equation centering as per template.
  4. All symbols and operators in equations should be explained before or immediately after they are entered. For example, operators E[] and ·T introduced in Equations (2) and (6) are not explained. Please check the remaining symbols and operators using in equations.
  5. Please use references to equations as, e.g., ‘Equation (1)’ instead of ‘Equation 1’. Please use ‘Equation’ instead of the ‘Eq.’ abbreviation (e.g., line 193, page 6). Please use ‘versus’ instead of the ‘vs.’ abbreviation (e.g., line 223, page 7).
  6. Use constant leading (spacing) throughout the document as recommended in the template.
  7. Please improve the description of the references according to the template. For this purpose, I suggest using a reference manager, e.g., Zotero: https://www.zotero.org/.
  1. English

In general, reading the article was not difficult for me. However, it is difficult for me to assess the linguistic correctness of the article in detail as English is not my national language. I think it is worth considering a detailed review of the article by a native speaker or a proofreading service. In addition, I recommend using the free application https://app.grammarly.com/.

A few sentences got my attention, e.g.,

  1. page 1, line 37: “… However, they involve phase …” maybe will be better instead of “…These, however, involve phase …”;
  2. page 6, line 186: “Input: B blocks of length N of a length-L noisy sinusoid …” is a little incomprehensible;
  3. page 8, lines 235-236: “…The larger the number of samples, the better the approximation is. …”.

Reviewer 2 Report

This paper used compressive covariance sensing (CCS) technology to estimate the frequency of single tone sinusoid in MA coloured noise, which is an interesting solution. However, this paper does not well address the following problems:

1) The presentation needs significant improvement. Both the paper presentation and organization need significant improvement. For example, line 172 – “the length-L sequence of noisy sinusoidal samples of Equation 1 is divided into B blocks of length N. Each block is compressed by the LSR of compression ratio M/N, and CCS is carried out” should be lifted to the start of section 2.

2) It is not very clear what the benefit of the proposed solution is. The CCS can only reduce 2/11 samples for processing, which might be trivial for most DSP applications. Also there have a branch of CS-based methods designed for DPD which can reduce the sampling rate to 90%. Although they are in different areas but it is worthy to check.

3) It is not clear how the proposed solution relates to coloured noise. Although in the result session, an LPF is used to colour the noise. The algorithm itself is not related to coloured noise. In the case that the paper would claim the algorithm is resilient to coloured noises, more experiments are required.

4) In Section 4, what is the default (fixed interval) sampling rate? The accuracies of different sampling rates will be different using the same signal/noise configuration.

5) The paper lacks of comparative studies to literature CS-based frequency estimation. Like “Estimation of Frequency of a Sinusoid from Compressive Sensing Measurements”

Reviewer 3 Report

This applied the compressive covariance sensing to estimate frequency of a tone from sparse measurements. The proposed method is interesting and should be useful in some cases. It can be considered for publication after some revisions.

The specific comments are in the following:

  1. What is 'MA' in the title of the paper? It should be removed.
  2. In lines 73 and 74, you mentioned "restricting the treatment to  moving-average (MA) noise and achieving low complexity". The reason of using the MA noise model should be explained more here.
  3. Only simulation results are presented in this paper. The authors can consider to add some experiment results. 

Round 2

Reviewer 1 Report

Thank you for carefully analyzing my review and considering all comments and suggestions in detail. 
Congratulations on a very interesting paper!!!

When editing the final version, I suggest correcting the wo symbol in graphs' axes in Figs. 6 and 7.